# Noble gases confirm plume-related mantle degassing beneath Southern Africa

S.M.V. Gilfillan [1]*, D. Györe [2], S. Flude[1,7], G. Johnson [3], C.E. Bond [4], N. Hicks[5], R. Lister[6], D.G. Jones[6], Y. Kremer [3], R.S. Haszeldine[1] & F.M. Stuart[2]

Southern Africa is characterised by unusually elevated topography and abnormal heat flow. This can be explained by thermal perturbation of the mantle, but the origin of this is unclear. Geophysics has not detected a thermal anomaly in the upper mantle and there is no geo-chemical evidence of an asthenosphere mantle contribution to the Cenozoic volcanic record of the region. Here we show that natural $CO_2$ seeps along the Ntlakwe-Bongwan fault within KwaZulu-Natal, South Africa, have C-He isotope systematics that support an origin from degassing mantle melts. Neon isotopes indicate that the melts originate from a deep mantle source that is similar to the mantle plume beneath Réunion, rather than the convecting upper mantle or sub-continental lithosphere. This confirms the existence of the Quathlamba mantle plume and importantly provides the first evidence in support of upwelling deep mantle beneath Southern Africa, helping to explain the regions elevation and abnormal heat flow.

[1] School of GeoSciences, University of Edinburgh, Grant Institute, James Hutton Road, Edinburgh EH9 3FE, UK. [2] Isotope Geosciences Unit, Scottish Universities Environmental Research Centre, East Kilbride G75 0QF, UK. [3] Department of Civil and Environmental Engineering, University of Strathclyde, James Weir Building, Glasgow G1 1XJ, UK. [4] School of Geosciences, University of Aberdeen, Meston Building, Kings College, Aberdeen AB24 3UE, UK. [5] Council for Geoscience, 139 Jabu Ndlovu St., Pietermaritzburg, KwaZulu-Natal 3200, South Africa. [6] British Geological Survey, Environmental Science Centre, Nicker Hill, Keyworth, Nottingham NG12 5GG, UK. [7] Present address: Department of Earth Sciences, University of Oxford, 3 South Parks Rd, Oxford OX1 3AN, UK. *email: stuart.gilfillan@ed.ac.uk

A striking feature of the African continent is the ~1 km elevation of the eastern and southern African plateaus. This has been termed the African Superswell[1], and is also manifest by the shallow bathymetry of the southeastern Atlantic Ocean basin[2]. Seismic imaging beneath the African continent has revealed a large anomalous zone of low seismic velocity[3], similar to that identified beneath the Pacific[4]. Termed Large Low Shear Wave Velocity Provinces (LLSVP) or superplumes, these are imaged to extend upwards from the core-mantle boundary[5]. Mantle flow induced by these velocity anomalies has been proposed to dynamically support elevated crustal regions[6]. The high topography of the eastern African plateau and unusual bathymetry of the southeastern Atlantic basin has been attributed to recent thermal modification of the upper mantle associated with the East African Rift System[7]. Recent geophysical[2] and geochemical[8] studies have indicated that the deeply rooted African superplume is the primary cause of this mantle anomaly, and is a major contribution to the Cenozoic rifting and volcanism of eastern Africa.

However, it is currently unclear if the anomalous topography of southern Africa is supported by a thermal perturbation in the lithospheric mantle[9], the sub-lithospheric upper mantle[10], the lower mantle[11], or a combination of all three[2]. Previous seismic studies of the upper mantle structure beneath southern Africa have recorded only a small decrease in seismic velocities within the sub-lithosphere mantle, indicating that a thermal anomaly is unlikely[2]. Alternative hypotheses for the uplift of the region include; heating of the lithosphere by the tail of a Mesozoic plume that was stationary beneath the area for more than 25 million years[9], or that it is the result of buoyancy from the African superplume present in the lower mantle[11].

The Lesotho-KwaZulu-Natal region exhibits the highest relief in southern Africa[12] forming the southernmost part of the African Superswell. The region experiences active seismicity[13] and the sedimentary record of the Durban Basin and other Cretaceous basins surrounding southern Africa provide evidence for deposits sourced from distinct pulses of uplift and erosion in the Turonian, Oligocene, mid-Miocene and Pliocene[14]. Offshore, the anomalous bathymetry[15] and seamounts[13] of the Mozambique Basin, have been linked to active mantle upwelling associated with the hypothesised Quathlamba mantle plume[13]. Onshore, this could also explain the seismicity[16], anomalous topography[12], small-scale volcanic activity[17], thermal springs[18], elevated geothermal gradient[19] and active $CO_2$ seeps[13,20] of the region.

However, the nature of the upwelling mantle and whether it originates in the deep or shallow mantle is not understood, nor is the relationship to the underlying African superplume. The isotopic composition of the noble gases (He, Ne and Ar) are an established geochemical method of distinguishing between deep undegassed[21,22] and shallow convective mantle sources[23]. The presence of a noble gas signature of the deep mantle source associated with the ongoing $CO_2$ degassing would provide a measure of whether mantle upwelling is related to the deep-sourced African superplume[24] as opposed to a shallow convection-driven process in the depleted upper mantle[8].

Here, we show that whilst the $^3He/^4He$ are lower than a typical primordial mantle source of >8 $R_A$ (where $R_A$ is the $^3He/^4He$ of atmospheric air of $1.399 \times 10^{-6}$), the Ne isotopic composition of the degassing mantle $CO_2$ requires a deep mantle source, similar to that tapped by intraplate volcanism at Réunion[25] or Kerguelen islands[26], rather than the convecting depleted upper mantle. This confirms the existence of the previously hypothesised Quathlamba mantle plume[13] and illustrates that even modest plume induced lithospheric mantle melting, which is yet to result in significant extrusive volcanism, has incorporated a noble gas signature of the deep mantle source. Our findings provide the first

geochemical verification of ongoing deep mantle upwelling in Southern Africa and corroborates existing geophysical evidence that small-scale mantle plumes are emanating from the top of the African LLSVP in the region[24].

## Results

**Natural $CO_2$ degassing in Lesotho-KwaZulu-Natal.** $CO_2$ gas seeps are common in areas of active or recent magmatism, and are frequently associated with fault-related fluid migration from depth[27]. Natural $CO_2$ degassing is rare in South Africa; the natural cold $CO_2$ seeps along the Ntlakwe-Bongwan fault in southern KwaZulu-Natal are the largest concentration of such phenomena. The fault was identified during geological mapping between 1911 and 1916[28] with the seeps first described in 1923[29] (Fig. 1a–c). The fault is expressed at the surface over ~80 km[30] and is defined by a ~70 km wide arcuate zone of faulting that evolves southwards from an ENE-WSW to a north-south strike[31]. It is believed to be related to Gondwana rifting[32] which commenced ~180 Ma and continues to the present (Fig. 1c).

The origin of the degassing $CO_2$ is enigmatic, with initial work proposing a link to dissolution of carbonate rocks at depth by acidic groundwater[33]. Later $\delta^{13}C$ and $\delta^{18}O$ measurements of the $CO_2$ indicated an origin from low temperature acidic groundwater reactions with carbonate rocks of similar composition to the Cambrian Matjies River Formation[20] of the Western Cape Province. Carbonates of the nearby Marble Delta Formation were ruled out as a source, as their $\delta^{13}C$ was found to be distinct from the $\delta^{13}C_{CO_2}$ measured in the exsolving $CO_2$[20], but this work did not take into account the potential fractionation of $\delta^{13}C$ that would result from formation of a free-phase $CO_2$ during dissolution of carbonate rock. Mantle melting associated with the hypothesised Quathlamba mantle hotspot has also been proposed as both a potential $CO_2$ source, and a cause of a local thermal anomaly[13].

Natural $CO_2$ can have multiple origins, including; shallow biogenic processes, carbonate hydrolysis, deep burial related mechanical breakdown or thermo-metamorphism of carbonates and degassing of magmatic bodies[34,35]. Whilst $\delta^{13}C_{CO_2}$ can often resolve these sources, $CO_2$-rich natural gases frequently exhibit values that overlap with the range of carbon from magmatic source and carbonate breakdown[34], making it challenging to resolve their origin[36]. Noble gas isotopes are powerful tracers of the origin of $CO_2$, particularly in identifying mantle contributions[37]. Primordial isotopes, such as $^3He$, originate in the Earth's mantle and gases from the depleted upper mantle define a narrow range of $CO_2/^3He$ (of 1 to $10 \times 10^9$)[38–40].

**Combining $\delta^{13}C_{CO_2}$ with $CO_2$ and helium measurements to resolve $CO_2$ origin.** Here we combine new noble gas analyses of Bongwan $CO_2$ and $\delta^{13}C_{CO_2}$ from six separate gas seeps, sampled from three locations along the Bongwan fault and associated splays (Supplementary Tables 1–2). $\delta^{13}C_{CO_2}$ range from −2.0 to −3.3‰ (V-PDB standard) in line with previous determinations[20]. $^3He/^4He$ ratios corrected for air ($^3He/^4He_c$—see "Methods") within the samples range from 3.6 to 4.5 $R_A$. These are considerably above the atmospheric ratio (1 $R_A$) indicating the presence of a significant amount of primordial $^3He$.

$^4He$ exhibits the widest range in concentration compared to the other noble gases, from $1.48 \times 10^{-9}$ (Umtamvuna Mound 2) to $9.62 \times 10^{-5}$ cm³(STP)cm⁻³ (Baker Farm). $^{20}Ne$ concentrations range from $4.01 \times 10^{-9}$ to $4.13 \times 10^{-8}$ cm³(STP)cm⁻³, with $^{40}Ar$ ranging from $6.34 \times 10^{-6}$ to $7.83 \times 10^{-5}$ cm³(STP)cm⁻³. As with $^4He$ concentrations, the lowest $^{20}Ne$ and $^{40}Ar$ values are exhibited by the $CO_2$ sampled from Umtamvuna Mound 2, and the highest values are from the Baker Farm sample. $CO_2$ from the Baker

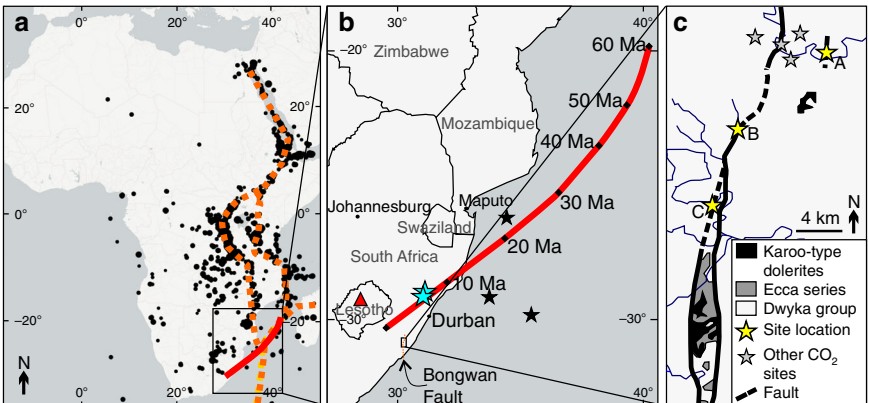

**Fig. 1** Location of study area and relevant geological features. **a** Map of the African continent illustrating depicting extent of panel 1b (inset box), the hypothesised track of the Quathlamba hotspot[13] (red line), trace of the East African Rift System (EARS)[64] (dashed orange lines) as picked out by earthquakes (black circles) between 2006 and 2016 (downloaded from the publicly available USGS database - https://earthquake.usgs.gov/earthquakes/search/ and plotted onto open source map tiles from the leaflet-extras R-library using Open Street Map tiles available via a CC-BY-SA licence from https://www.openstreetmap.org with country boundaries defined by Open Street Map). **b** Expanded map depicting study area (Panel c); the Bongwan fault trace (orange dashed line); the location of a small (1 m$^3$) basaltic eruption that took place in Lesotho in 1983[17] (red triangle); the locations of the Shu Shu and Lilani hot springs on the Tugela river (blue stars) which also exhibit minor $CO_2$ degassing and hot water (up to 53 °C)[65]; the position of anomalous seafloor mounds in the northern Natal Valley[54] (black stars) and the speculative track of the hotspot, based on African plate movement reconstructions, with corresponding proposed positions of the hotspot at the dates cited[13]. **c** Expanded map of Bongwan fault showing the location of the three sampling sites from this study marked as yellow stars (A – Baker Farm, B – Mjaja and C – Umtamvuna; location letters correspond to previous work[27]), the trace of the fault and the bedrock geology of the region. Historically reported[28,33], but no longer accessible or existing seep locations are depicted as grey stars

Farm and Mjaja seeps (A and B on Fig. 1c) exhibit $CO_2/^3He$ of 1.88 and $6.78 \times 10^9$, confirming a mantle origin (Fig. 2). The four seeps sampled at Umtamvuna exhibit considerably higher $CO_2/^3He$ ratios. As $^3He$ is inert and insoluble[37], and there is no significant $^3He$ in the crust[37] ($^3He/^4He_{crust} = 0.05$ $R_A$)[41], the variation in $CO_2/^3He$ is predominantly linked to the addition of $^3He$-poor $CO_2$[42].

Combining $CO_2/^3He$ with $\delta^{13}C_{CO2}$ allows organic sediment and limestone-derived $CO_2$ to be distinguished from magmatic sources[38] (Fig. 2). The trend between $CO_2/^3He$ and $\delta^{13}C_{CO2}$ are consistent with the mixing of mantle-derived $CO_2$ with $CO_2$ derived from the overlying Marble Delta Formation carbonates at up to 70 °C (see "Methods"). Based on the regional geothermal gradient of 30 °C/km[19] and an average surface temperature of 14 °C[43], we estimate that mixing occurred at depths of less than ~1900 m.

Linking $CO_2/^3He$, $CO_2/^4He$ and $^3He/^4He$ in a ternary plot allows $CO_2$ sources to be resolved[44,45], permitting direct comparison of the relative proportions of $CO_2$, $^3He$, and $^4He$, regardless of absolute concentrations (Fig. 3). Binary mixtures and loss or gain of a single component plot as straight lines on ternary plots. Figure 3 demonstrates that the Baker Farm gas requires the ingrowth/addition of 33 to 50 % radiogenic $^4He$, derived from the lithosphere, to mantle magmas, relative to the depleted upper asthenosphere mantle (DM) ($8 \pm 1$ $R_A$)[46] or sub-continental lithospheric mantle (SCLM) ($6.1 \pm 0.9$ $R_A$)[47], respectively. The remaining samples require the addition of $^3He$-poor $CO_2$. The modest $^4He$ required in this gas confirms a shallow crustal origin for this non-magmatic $CO_2$ (also see Supplementary Fig. 1). The low $^4He/^{20}Ne$ in these samples contrasts with the higher values measured in the Mjaja and Baker Farm gases and implies that the $CO_2$ has interacted with atmosphere-saturated groundwaters.

Recent work undertaken on $CO_2$ seeps in Australia, similar to those at Bongwan, has highlighted that equilibration of mantle-sourced $CO_2$ with atmosphere-saturated groundwaters, followed by solubility controlled fractionation during exsolution of the $CO_2$ from the groundwater at $CO_2$ seeps, can result in depleted

He concentrations and elevated $CO_2/^3He$ from values which are originally within the magmatic source range[48]. Given the wide range of $^4He$ concentrations observed between the different seeps, it is likely that both mixing with crustal derived $CO_2$ and solubility fractionation, resulting in a relative loss in He, have acted to produce the observed $CO_2/^3He$ ratios at Bongwan (Fig. 3).

**Constraining the mantle source using Ne and Ar isotopes**. The $^3He/^4He$ of the SCLM of the Karoo Large Igneous Province and the nearby volcanic ocean islands (Comores, Tristan da Cuna and Gough) are characterised by low $^3He/^4He$ (4.9–7.1 $R_A$)[49]. Hence, the mantle source in the region may have $^3He/^4He$ that is lower than MORB, but is most likely to be above the highest measured $^3He/^4He_c$ of 4.27 $R_A$. The $CO_2$ degassed at the Bongwan seeps has migrated from the mantle through the crust and will have incorporated radiogenic He from the Precambrian metamorphic basement and sedimentary cover. Radiogenic $^4He$ is required to account for $CO_2$-He isotope systematics of the Baker Farm seep gas (Fig. 3) and can account for the $^4He/^{21}Ne^*$ ratio of $1.23 \times 10^7$ in the sample, below the crustal ratio of $1.71 \times 10^7$ [41]. Neon isotopes provide less ambiguous insights into the mantle source, enabling differentiation of depleted, convecting upper mantle (DM), the source of mid-ocean ridge basalts (MORB) and the primordial $^{20}Ne$-enriched mantle that is sampled by intraplate magmatism, the source of ocean island basalts (OIB)[50].

The Ne isotope composition of Baker Farm and Mjaja seeps show a clear mantle component, which is distinct from both atmospheric Ne and the mass fractionation line (Fig. 4). $^{40}Ar/^{36}Ar$ of the Mjaja and Baker Farm gases ($550 \pm 2$ and $961 \pm 4$ respectively) are higher than the air value, consistent with a partial mantle origin of the non-atmospheric $^{40}Ar$. This is supported by the $^4He/^{40}Ar^*$ (1.55 and 1.77 respectively) which are close to the mantle value of 2[37]. The remaining samples have atmospheric dominated Ne and Ar isotope compositions consistent with derivation of Ne and Ar from air-equilibrated groundwater. This corresponds to the crustal $CO_2$ addition from

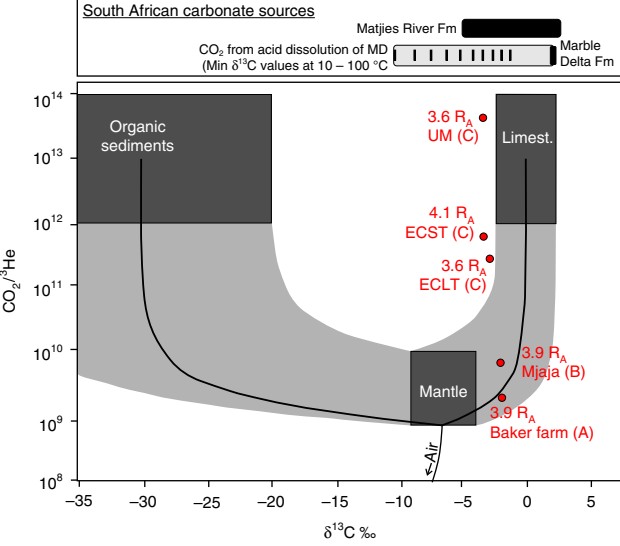

**Fig. 2** $CO_2/^3He$ against $\delta^{13}C$ for the Bongwan $CO_2$ samples. Individual samples are plotted as red circles along with labels outlining their associated air-corrected $^3He/^4He_c$ in $R_A$ (where $R_A$ is the atmospheric $^3He/^4He$ of $1.399 \times 10^{-6}$) and their location corresponding to Fig. 1 (A – Baker Farm, B – Mjaja and C – Umtamvuna). Mixing lines (black lines and grey shading) are shown for $CO_2$ derived from the mantle, limestone and organic sediments[38]. End-member compositions cover the range of values[38,62] (mantle $CO_2/^3He = 1-10 \times 10^9$; mantle $\delta^{13}C = -9$ to $-4\%o$; Crustal $CO_2/^3He = 1 \times 10^{12}-10^{14}$; Limestone $\delta^{13}C = 0 \pm 2\%o$; Organic sediment $\delta^{13}C = -30 \pm 10\%o$). The range of $\delta^{13}C$ values for South African carbonate sources from the Marble Delta[20], Matjies River Formation[20], and the Transvaal Supergroup[66] are provided. The extent of $\delta^{13}C_{CO_2}$ that would be produced by acid groundwater dissolution of Marble Delta Formation carbonate is also shown (grey box and black lines depicting the minimum $\delta^{13}C_{CO_2}$ that would result from dissolution of the formation carbonate between 10 °C (left) and 100 °C (right)) (see Methods). The trend between $CO_2/^3He$ and $\delta^{13}C_{CO_2}$ are consistent with the mixing of mantle-derived $CO_2$ with $CO_2$ derived from the overlying Marble Delta Formation carbonates at up to 70 °C (see "Methods"). $\delta^{13}C_{CO_2}$ was not measured from the sample collected at the Umtamvuna River Spring and hence this sample cannot be depicted on the plot

groundwater and/or He loss due to degassing of $CO_2$ from groundwater[48], required to account for the elevated $CO_2/^3He$ in these samples, compared to the more mantle-rich values of Mjaja and Baker Farm.

Importantly, the results of the high precision Ne analysis of Mjaja and Baker Farm seep gases do not plot on the MORB-air mixing line in Ne isotope space (Fig. 4). Instead they provide a clear indication that the mantle source of the $CO_2$ is more primordial than that of the convecting upper mantle (Fig. 4). The duplicate high precision determinations of the Baker Farm and Mjaja $CO_2$ samples overlaps with the trend defined by the Kerguelen[26] and Réunion hotspots[25], implying the ultimate source of the upwelling mantle is deep. The low $^3He/^4He$ of the Bongwan gases relative to Kerguelen (12.3 ± 0.3 $R_A$) and Réunion (11.5–13.1 $R_A$) would require the incorporation of crustal radiogenic $^4He$.

However, the $^3He/^4He$ of the sub-lithospheric mantle source of the nearby Karoo Large Igneous Province is cited as 7.03 ± 0.23 $R_A$[51] and the most proximal volcanic ocean islands (Comores, Tristan da Cuna and Gough) are characterised by low $^3He/^4He$ (4.9 to 7.1 $R_A$)[51]. Hence, the mantle source in the region may have a $^3He/^4He$ that is not elevated above the MORB range (8 ± 1 $R_A$), though is most likely to be above the highest measured $^3He/$

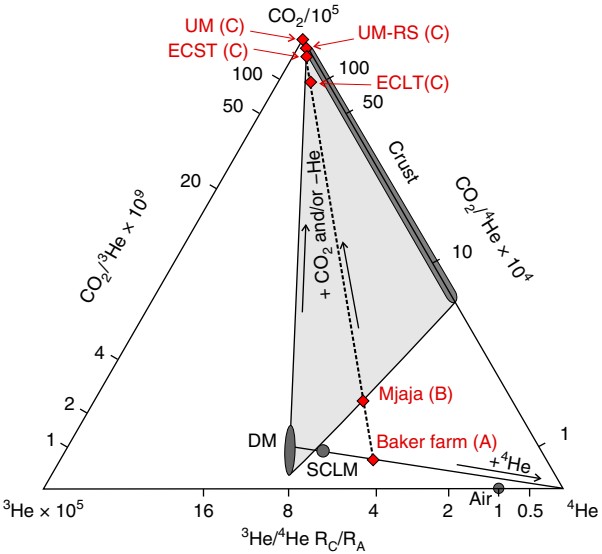

**Fig. 3** Ternary plot of $CO_2/^3He$, $CO_2/^4He$ and $^3He/^4He$ of Bongwan $CO_2$ samples. Relevant end-members, after[45], with relative concentrations calculated for $CO_2$, $^3He$, and $^4He$, based on these ratios. Scaling factors are introduced to the $CO_2$ and $^3He$ values, as labelled, to allow the full spread of data to be represented. The scaled relative concentrations are then normalised to a three-component system where $CO_2 + ^3He + ^4He = 100\%$, permitting direct comparison of the relative proportions of $CO_2$, $^3He$, and $^4He$, regardless of absolute concentrations. Binary mixtures and loss or gain of any single component will plot as straight lines. The plot shows that the Bongwan $CO_2$ samples have been influenced by two separate processes: (1) Mixing between depleted upper mantle (DM) (8 ± 1 $R_A$)[46] or sub-continental lithospheric mantle (SCLM) (6.1 ± 0.9 $R_A$)[47] and a radiogenic $^4He$ component, and (2) Addition of a high-$CO_2$, low-$^3He$ and $^4He$ component and/or loss of both $^3He$ and $^4He$ relative to $CO_2$. The Baker Farm sample lies on the first mixing line, and the addition of $CO_2$ free of $^3He$ and low in $^4He$ and the effect of shallow preferential degassing of He relative to $CO_2$ in the near-surface[48] can account for the second trend. ECLT – East Cape Large Travertine, ECST – East Cape Small Travertine, UM-RS – Umtamvuna River Spring and UM – Umtamvuna

$^4He_c$ of 4.5 $R_A$. It is also possible that the He and Ne systematics of the mantle under southern South Africa are decoupled, as has been observed in the Icelandic and the Colorado Plateau mantle sources[23,52,53]. This decoupling was attributed to either more compatible behaviour of He during low-degree partial melting or more extensive diffusive loss of He relative to the heavier noble gases. Incorporation of crustal-radiogenic $^{21}Ne$ to the Bongwan gases is also probable, but without constraint of the original Bongwan mantle $^3He/^4He$ this is impossible to determine.

## Discussion

The Bongwan $CO_2$ seeps are located at the end of the hypothesised Quathlamba hotspot track. Hotspot migration has been proposed to explain chain of volcanic seamounts that track across the Mozambique Basin[13], orientated in a direction that closely resembles reconstructions of the African plate movement (Fig. 1). Recently, anomalous 30 km elongate seamounts, have been identified within the Northern Natal Valley offshore of Durban[54]. The geospatial positioning of these could extend the East African Rift System southwards, but they are also within range and age of the proposed Quathlamba hotspot track (Fig. 1b). Furthermore, recent work has found that the seafloor adjacent to the Mozambican continental margin, and that of the central Mozambique Channel is 300 m and 1300 m shallower,

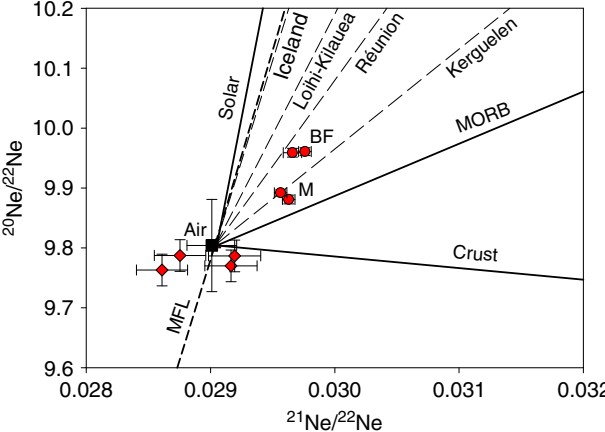

**Fig. 4** $^{20}$Ne/$^{22}$Ne plotted against $^{21}$Ne/$^{22}$Ne for the Bongwan $CO_2$ samples. Lower precision analysis performed on the MAP 215-50 mass spectrometer are plotted as red diamonds, higher precision analysis undertaken on the ARGUS VI mass spectrometer are depicted as red circles, with air plotted as a black square. 1σ errors associated with each measurement are provided along with mixing lines between air[67], continental crust[41], MORB[46] and solar end members[62] shown as solid black lines. The Kerguelen, Reunion, Loihi-Kilauea, and Iceland hotspots are depicted as thin dashed lines[46] with the mass fractionation line (MFL) shown as a thick dashed line. The Baker Farm (BF) and Mjaja (M) gases plot above the established air-MORB mixing line[40,68], plotting between the Kerguelen and the Reunion mantle source[25], implying an undegassed mantle origin for the Bongwan $CO_2$. The uncertainty differences reflect different analytical procedures and the ARGUS samples have been corrected for the contributions to $^{21}$Ne from NeH$^+$, which is controlled by the amount of $H_2$ present in the mass spectrometer during analysis[61] (see "Methods")

respectively, than the conjugate basins in Antarctica, or than oceanic thermal subsidence models predict[15]. This has been attributed to the presence of thickened oceanic crust, linked to the passage of a mantle plume beneath the basin during the Paleogene[15].

Plate movement reconstructions indicate that this hotspot moved under the continent approximately 10 million years ago (Fig. 1), coinciding with several periods of regional uplift[55]. The Ne isotope systematics of the $CO_2$-rich gas seeps provide the first geochemical evidence that small volumes of melting is occurring beneath the continental lithosphere in the region. This confirms the existence of the previously hypothesised Quathlamba mantle plume[13] and illustrates that even modest plume induced lithospheric mantle melting, which is yet to result in significant extrusive volcanism, has incorporated a noble gas signature of the deep mantle source. These findings provide the first geochemical verification of ongoing mantle upwelling in Southern Africa, confirming geophysical evidence that small-scale mantle plumes are emanating from the top of the African superplume[24]. Buoyant underplating of Southern Africa by the African superplume provides an explanation for the anomalous elevation, high heat flow, and how the Quathlamba mantle plume has incorporated deep-sourced mantle volatiles.

## Methods

**Fieldwork**. The $CO_2$ seeps were identified in the field as bubble streams in pools of water, rivers and wellbores (Fig. 1, Supplementary Table 1). Gas samples were collected in September 2015 by placing a plastic funnel over the site of the $CO_2$ seep and flowing the gas through a 70 cm length of refrigeration grade copper tubing fitted with an exhaust hose to prevent turbulent back-mixing of air into the sample. The tubing was purged with the seeping gas for between 10 and 15 min before being clamped by a purpose built tube clamp at both ends to seal the copper tube

with a cold-weld that is impervious to helium[36]. Tedlar sample bags were filled at each seep for stable isotope analyses. The Baker Farm borehole was sampled by sealing the well and using a soil gas probe to collect gas from as deep as possible within the well. A Geotechnical Instruments GA2000 portable gas analyser was then used to extract gas from the Baker Farm well, with the pump being connected downstream of the sampling apparatus. Further details on individual field sites and other surveys undertaken in the area are outlined in Supplementary Table 1 and in previous work[27,56].

**Laboratory analysis**. Bulk gas, stable isotope and noble gas analysis was undertaken at the Scottish Universities Environmental Research Centre (SUERC), using previously described techniques[57]. Bulk gas content as a percentage was determined using a Pfeiffer Vacuum QMS 200 quadrupole mass spectrometer with all seeps sampled exhibiting concentrations of >99% $CO_2$. $\delta^{13}C_{CO_2}$ were measured using a VG SIRA II dual inlet isotope ratio mass spectrometer following established procedures[58]. Precision and reproducibility are typically better than ±0.2‰ for $\delta^{13}C$ (Supplementary Table 2).

Noble gas analyses from all samples were performed on volumes of ~10 cm$^3$ gas stored in copper tubes. Each sample was expanded to a titanium sublimation pump (900 °C) and a series of SAES GP50 ZrAl getters (250 °C) operating under ultra-high vacuum, following established procedures[57–60]. The isotopic composition of He, Ne and Ar of all six samples was measured using a MAP 215-50 mass spectrometer using established techniques[57–60] (Supplementary Tables 2 and 3). Analytical errors are governed by the reproducibility of air calibrations, for Ne, standard reproducibility was assessed using the best Gaussian fit to the probability density distribution of $^{21}$Ne/$^{22}$Ne and $^{20}$Ne/$^{22}$Ne ratios from 14 air calibrations, which is an objective way of filtering outliers. These samples were not corrected for any $^{20}$NeH$^+$ contribution to $^{21}$Ne.

High precision analysis of Ne isotopes within the Baker Farm and Mjaja samples was undertaken in multi-collection mode using a ThermoFisher ARGUS VI using the following procedures. Each copper tube sample was mounted on the ultra-high vacuum line attached to the MAP 215-50 mass spectrometer, and subjected to the same clean up procedure as outlined above, following which they were trapped in a 2 L stainless steel cylinder. Approximately 100 cm$^3$ of total gas was extracted from this cylinder to the ultra-high vacuum system attached to the ARGUS VI mass spectrometer as described in previous work[61]. The gas was exposed to another SAES GP50 ZrAl getter (held at 250 °C) for 15 min and then a liquid nitrogen-cooled charcoal finger (held at −196 °C) for 15 min to trap any remaining active gases along with Ar, Kr and Xe. Ne was then adsorbed on charcoal using an IceOxford cryopump (−243 °C, 20 min) while He was pumped away. Pure Ne was released at −173 °C and administered into the ARGUS VI low resolution mass spectrometer. The ARGUS clean-up and analysis procedure was undertaken twice for both the Baker Farm and Mjaja samples, and the results of the individual repeat measurements are plotted on both Fig. 4 and Supplementary Fig. 3, and listed in Supplementary Table 4.

Analysis of Ne isotopes followed procedures described in previous work[61]. Ne isotopes were multi-collected ($^{22}$Ne$^+$– H$_2$, $^{21}$Ne$^+$ – Axial, $^{20}$Ne$^+$ – L2) on 10$^{12}$ Ω Faraday amplifiers. Isobaric interferences of $^{40}$Ar$^{2+}$ and $^{44}$CO$_2^{2+}$ were quantified using pre-determined singly/doubly charged ratios under measurement conditions and the in situ measurement of $^{40}$Ar$^{2+}$ and $^{44}$CO$_2^{2+}$ during analysis on the CDD detector. The contributions of $^{40}$Ar$^{2+}$ to the corresponding $^{20}$Ne peak were found to be ~0.6% and 0.07% for Baker Farm and Mjaja samples, respectively. Contributions of $^{44}$CO$_2^{2+}$ to $^{22}$Ne for Baker Farm and Mjaja were found to be 0.2% and 0.04%, respectively. The contribution of $^{40}$Ar$^+$ and $^{44}$CO$_2^+$ to the overall uncertainty of Ne isotopic ratios was found to be below 0.01%. Other interferences (H$_2$$^{18}$O$^+$, H$^{19}$F$^+$, $^{65}$Cu$^{3+}$) were found to be negligible. The $^{20}$NeH$^+$ contribution at m/z = 21 was determined using a pre-recorded calibration curve of $^{22}$Ne – $^{22}$NeH at a constant hydrogen level, that exceeded the level of $^{22}$Ne, where $^{20}$Ne of each sample was measured[61]. The measurement of $^{22}$NeH occurred at $m/z = 23$, corrected for $^{46}$CO$_2^{2+}$ and blank. $^{20}$NeH$^+$ correction at m/z = 21 was found to be 0.96% (Baker Farm) and 1.94% (Mjaja). The contribution of NeH correction toward the overall uncertainty is ± 0.02%. Mass fractionation was corrected by the repeated analysis of air prior to and after analysis ($n = 9$) with the reproducibility of $^{20}$Ne/$^{22}$Ne = 0.05% and $^{21}$Ne/$^{22}$Ne = 0.11%.

**Data analysis**. $^3$He/$^4$He were corrected for minor atmospheric air contributions using the measured $^4$He/$^{20}$Ne, following the established formula:[37]

$$\left({}^3He/{}^4He\right)_c = \frac{\left({}^3He/{}^4He\right)_s \times \left({}^4He/{}^{20}Ne\right)_s / \left({}^4He/{}^{20}Ne\right)_{air} - \left({}^3He/{}^4He\right)_{air}}{\left({}^3He/{}^{20}Ne\right)_s / \left({}^3He/{}^{20}Ne\right)_{air} - 1} \quad (1)$$

Mixing curves shown in Fig. 2 are calculated after[62] using E

$$\left(\frac{{}^{13}C}{{}^{12}C}\right)_{Obs} = A * \left(\frac{{}^{13}C}{{}^{12}C}\right)_A + B * \left(\frac{{}^{13}C}{{}^{12}C}\right)_B + C * \left(\frac{{}^{13}C}{{}^{12}C}\right)_C \quad (2)$$

and

$$1/\left(\frac{CO_2}{{}^3He}\right)_{Obs} = A/\left(\frac{CO_2}{{}^3He}\right)_A + B/\left(\frac{CO_2}{{}^3He}\right)_B + C/\left(\frac{CO_2}{{}^3He}\right)_C \quad (3)$$

where A, B and C refer to three different components and A+B+C = 1.

The predicted $\delta^{13}C_{CO_2}$ produced by the acid dissolution of the Marble Delta Formation between temperatures of 10 and 100 °C, depicted on Fig. 2, was calculated from the measured Marble Delta Formation $\delta^{13}C_{carb}$[20] using established fractionation factors between $\delta^{13}C_{carb}$ and gaseous $CO_2$, calculated according to equation [4][63], where $T$ is the temperature in Kelvin:

$$10^3 ln\alpha = -2.988 * (10^6/T^2) + 7.6663 * (10^3/T) - 2.4612 \qquad (4)$$

## Data availability

All data used to generate the figures are provided in Supplementary Tables 1–4, and Supplementary Figs. 1–3.

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

## Acknowledgements

S.M.V.G., S.F. and F.M.S. were supported by EPSRC Grant EP/K036033/1; C.E.B. was supported by NERC Grant NE/M007251/1; G.J. was supported by EPSRC Grant EP/P026214/1; R.S.H. was supported by the Scottish Funding Council, EPSRC Grants EP/P026214/1, EP/K000446/2 and NERC Grant NE/L008475/1. The authors would like to acknowledge the financial support of the UK CCS Research Centre (UKCCSRC) to undertake the sampling trip to South Africa. The UKCCSRC is funded by the EPSRC as part of the RCUK Energy Programme. The South African National Energy Development Institute (SANEDI) Stakeholder Engagement team under the South African Centre for Carbon Capture & Storage (SACCCS) are thanked for making the scientific work possible. The National, Provincial and Local Government structures including Traditional Authorities, Municipalities, landowners and local residents are thanked for granting permission to conduct the sampling of the seeps in the areas of interest. Council for Geoscience staff are thanked for their assistance and support in the field. Terry Donnelly and Marta Zurakowska are thanked for their assistance with stable isotope and noble gas analyses, respectively.

## Author contributions

N.H. introduced the authors to the study site. S.M.V.G., R.S.H. and F.M.S. designed the study. C.B., D.G.J., N.H. G.J. Y.K. and R.L. assisted with sample collection and background geological information. S.F. and D.G. analysed the first set of samples and performed initial data interpretation with assistance from F.M.S. D.G. and F.M.S. undertook the additional high precision analysis of Baker Farm and Mjaja samples. S.M.V.G., D.G., S.F. and F.M.S. interpreted the data and wrote the paper with input from all co-authors.

## Competing interests

The authors declare no competing interests.
