## [Peer Review File · Nature Communications]

Reviewers' comments:

Reviewer #2 (Remarks to the Author):

Review of Gilfillan et al.

This manuscript reports evidence for the presence of a mantle plume beneath South Africa in the Quathlamba region based on noble gases. The $^3\text{He}/^4\text{He}$ evidence falls short of proving that the source is of a deep mantle origin. However, the Ne isotopes may provide a better evidence as the Ne isotope signature corrected for mixing with a crustal component is similar to the Ne signature of the Kerguelen hotspot. This may be a solid enough evidence for a mantle plume. The key issue in evaluating this manuscript is whether the corrected Ne isotope data is indeed robust and whether this correction is valid. If this is the case, then the findings of this study may be interest to the geoscience community. On a more general note, the application of Ne isotopes to identifying the origin of mantle plume is not novel and has been widely used previously. Thus, I don't think the approach used in this study is novel enough and it would fit better in a more focused journal.

Validity of the Ne correction :

First, the authors have given some equations for the $^3\text{He}/^4\text{He}$ $\text{CO}_2/^3\text{He}$ isotope data to identify various components but do not explain properly how the correction is done for Ne isotopes. This should be part of the manuscript. Why is the minimum contribution from radiogenic derived ^{21}Ne estimated by taking the difference between SCLM and the least radiogenic helium sample? This seems to make the assumption that there is a component with a $^3\text{He}/^4\text{He}$ ratio higher than 4.25 Ra in the mixture but are we certain this is the case ? If the ratio of the other component is lower than that of average SCLM, what happens to the corrected Ne isotope ? There are some reservoirs including mantle plumes characterized by low $^3\text{He}/^4\text{He}$. By making this assumption, the authors may have automatically produced a Ne reservoir with a $^{20}\text{Ne}/^{22}\text{Ne}$ versus $^{21}\text{Ne}/^{22}\text{Ne}$ signature typical of mantle plumes. In the absence of more detailed explanations, it seems that the corrected Ne data may not be so robust.

Second, there should be a propagation of error calculations for the corrected Ne data. The data shown in Figure 4a has relatively large errors and it is not clear whether propagation of errors has been done. There are uncertainties in the $^4\text{He}/^{21}\text{Ne}$ ratios due to crustal production as these ratios depend on cross sections of neutron reactions with ^{24}Mg . Similarly the $^3\text{He}/^4\text{He}$ value of the SCLM has a typical variability of approximately 1 Ra.

Line 42 : Edge driven convection instead of edge driven convention

Same in line 48 'mantle convention'

Line 57 Rifting and mantle plumes could be both related to each other. This idea has been previously proposed in the literature.

Line 68 What is a moderately undegassed mantle ? I could understand the concept of a moderately degassed mantle but what a moderately undegassed mantle means is more obscure.

Line 117 correct 'primoidal ^3He '

Figure 3 : The ternary diagram used for representing the data in Figure is not a classical tool of noble gas geochemistry. The coordinates that are used are non linear and the components do not sum up to 100%. It would be good to explain perhaps in the supplement how this is built and how the coordinates are calculated. There is no sign of a mantle plume component in this diagram and this should be explained.

The discussion after line 214 is only a review of previous work and is not specifically original to this study. It is true that the presence of a mantle plume beneath south Africa forces one to reinterpret the observations made in this area. Yet the hypothesis of a mantle is not novel, which means this part of the interpretations does not make this study stronger. One other weakness is that the noble gas signature was only measured in gas seeps and therefore the connection with a mantle plume cannot be made for magmatic rocks directly.

Reviewer #3 (Remarks to the Author):

This paper reports the first noble gas data from the Natal province in SA. Because these elements are powerful tracers of mantle-crust interaction and mantle domain provenance, this study is welcome to document an important geodynamical problem from a different perspective. The region presents several indications of recent tectonic and volcanic activities, and current models for this province advocate either the interaction of a mantle plume source with the cratonic lithosphere, and/or regional rifting.

The present data are not many, supposedly because the gas manifestations are scarce, and the authors did a good job in documenting the composition of several key geochemical tracers. They show without ambiguity the contribution of a mantle source for these CO_2 -rich manifestations, which could not be demonstrated clearly with only stable isotope and chemistry data. The $\text{C}/^3\text{He}$ ratios of the samples with elevated $^{40}\text{Ar}/^{36}\text{Ar}$ ratios are within the range of mantle values, and the $^3\text{He}/^4\text{He}$ ratios values up to $4R_a$ cannot be a crustal signature.

In a further step, the authors argue that the mantle source is different from the convective mantle feeding mid-ocean ridges because the Ne isotope ratios lie above the air-MORB mixing line. Here the message is more ambiguous, because only one sample is marginally above the correlation, at 1 sigma level. Such measurements are generally difficult, but with modern instrumentation and the large quantity of sample allowed by natural gases, it should be possible to reach a better analytical precision and give the data at the 2 sigma level, potentially resolving this ambiguity.

The authors then correct the data with the following assumptions: noble gases were contaminated by a crustal component, so that the He isotope ratio of $4R_a$ results from mixing of crustal radiogenic 4He with a local mantle component having a value within $4.9\text{-}7.1R_a$, based on data from the Karoo large igneous province and on "nearby" volcanic islands (Comores, Tristan, Gough). I guess the concept of nearby may have a different flavour at the African continent scale, but the justification of this mantle end-member signature appears disputable. The authors argue that this is consistent with a subcontinental lithospheric mantle signature, in which case this appears difficult to reconcile with a less degassed mantle plume origin. Accepting this mantle He isotope signature, the authors propose to correct the Ne data for crustal contamination using the $4\text{He}/^{21}\text{Ne}$ radiogenic production (or accumulation, not specified) ratio published in the literature. The ms or its supplementary should absolutely state the assumptions, detail the calculations, give the mixing equations for the 2 He isotopes and the 3 neon isotopes, and give the propagation of analytical errors and uncertainties on end-member compositions. Another concern is the assumption that crustal 4He and ^{21}Ne are not fractionated during crustal contribution, which remains to be demonstrated. The authors could make some tests demonstrating the effect of potential fractionation on the corrected Ne isotope ratios.

Reviewer #4 (Remarks to the Author):

Gilfiligan et al. - Upwelling of the Quathalamba mantle plume beneath Southern Africa

This study uses noble gases to identify a mantle component in CO_2 gases collected from seeps in Southern Africa. The conclusions are justified and I believe the work is of wide interest. It is generally well written. I have only have minor comments some of which are stylistic. I recommend it is suitable for publication in Nature Communications with only minor modifications.

1)

The intro and abstract could be improved by stating more simply what observations need to be explained, why it is important and what has prevented progress previously. The last of these 3 ingredients is currently missing.

For example, the first line states 'The effect of upwelling mantle plumes on initiating uplift of cratonic continental crust and generating intraplate volcanism is poorly understood.'

This is too complex! There is a lot to unpack in that sentence which has already invoked mantle plumes. Nothing in the new data actually prove a link between plumes and initiation of uplift, so it is not all that pertinent.

I think something simpler along the lines of 'Mantle plumes are sometimes invoked to explain the anomalously high topography and heatflow of southern Africa. However, geochemical evidence for the presence of a mantle plume or the nature of the mantle reservoir sampled has been lacking because....? (inaccessibility of volcanic products? ambiguous radiogenic isotope signatures? difficulting of attributing radiogenic isotopes to mantle source versus crustal contamination? why is the involvement of a plume ambiguous - I would like to know?). Here we use noble gases in CO₂ seeps to overcome these difficulties and show.....

2)

Some of the language used to convey the significance of the work is a tad strong for my taste e.g. on line 19 - 'uniquely show an unambiguous..' the next sentence goes on to suggest the data characterise a primitive plume component rather than depleted mantle. The language may be technically correct, but the sentiment conveyed is a little strong. The evidence for mantle involvement is compelling, but the plume signature is not as compelling. I think it would be fairer to use words like 'favour a plume component' or are 'consistent with....'..a source similar to the Kerguelen source'.

3)

Line 113 and thereabouts. What is the concentration of noble gases within the CO₂ gases?

4)

I find Figure 3 confusing. This kind of trigonal plot is usually used to separate 3 end-members, in this case CO₂, ³He, ⁴He. However, the area shown represents a small fraction of the complete triangle - in this case the bottom axis still has 10,000 times more CO₂ than ⁴He (or put another way, the apices should be labelled as mixtures of gas, not pure ³He and ⁴He).

I am also unsure if the reference fields or axes are correctly labelled.

Air is placed on the bottom axis at ³He/⁴He of 1 Ra. This should be a CO₂/⁴He of ~80 because air contains ~5ppm ⁴He and ~400 ppm CO₂. It is difficult to read the CO₂/⁴He from the axis which has been raised x10,000, however, 80 divided by 10,000 would be 0.0080.

Silicate rocks frequently contain hundreds of ppm range of CO₂ but are generally strongly enriched in ⁴He compared to air and also the mantle. Therefore crust should surely have a CO₂/⁴He of less than air and less than the mantle?

The field labelled as crust might represent a specific unit or carbonate sediments, but I cannot foresee it is representative of the whole crust.

As a result of the chosen fields one of the data points lies outside the apparently allowed area of mixing (shaded grey) and is plotted on a vector toward pure ⁴He. However, surely this vector is toward crust? (It's not pure ⁴He given the apex is a ⁴He-rich mixture....)

The data appear to define a pseudo-binary array. I think the first end-member - Baker Farm (A) is a mixture of mantle and deep crustal gases (e.g. ancient silicates) and the 2nd endmember near the CO₂ apex is a ground water or carbonate-related component. This is similar to what is written, but I cannot reconcile what is written with the diagram - sorry!

Can the range of CO₂/⁴He in groundwater be plotted on this figure?

5)

lines 168-169 - air equilibrated groundwater is mentioned here as a component so should be shown in Fig 3.

6)

Fig 4 is also confusing. There are six blue data points and 2 red ones. The red ones are discussed as being the measured values of the two samples with mantle Ne A and B (ref to Fig 3) and the blue points labelled A' and B' are corrected for radiogenic ingrowth.

Presumably the other blue points correspond to the measured values of the gases labelled C in Fig 3, but this needs to be clarified. Why aren't corresponding red points shown for these analyses? because there is no correction probably, but then why are the points blue instead of red?

The best fit lines are not clear. Presumably they are forced through air? but this isn't stated. I don't think they are needed as we can already see the blue points lie close to the Kerguelen reference line.

Where does Ne from the Karoo province or Tristan de Cunha plot in this diagram? this would be more relevant than Kerguelen, as you mention it is a possible composition for background mantle when discussing Ne.

Fig 4b doesn't really tell us anything useful. The slopes of the curves (not lines) are related to ^{36}Ar and ^{22}Ne concentrations in the end-members and could be altered by a number of processes not related to source composition. I would recommend moving this to the supplement as it doesn't help the nice story.

7)

Methods. I appreciate space restrictions and that the methods have been described before and SUERC are experts in these techniques, but I think you should at least specify the instruments used for the measurements here. It wasn't all done on a quadrupole!!

Mark Kendrick

Response to reviewers for Gilfillan et al., Upwelling of the Quathlamba mantle plume beneath southern South Africa

Key to document

Reviewers' Comments: Plain text, Calibri font

Authors Responses: *Italic, indented text, Calibri font*

Quoted Revised Text: Plain indented text, Times New Roman font

Reviewer #2 (Remarks to the Author):

Review of Gilfillan et al.

This manuscript reports evidence for the presence of a mantle plume beneath South Africa in the Quathlamba region based on noble gases. The $3\text{He}/4\text{He}$ evidence falls short of proving that the source is of a deep mantle origin. However, the Ne isotopes may provide a better evidence as the Ne isotope signature corrected for mixing with a crustal component is similar to the Ne signature of the Kerguelen hotspot. This may be a solid enough evidence for a mantle plume. The key issue in evaluating this manuscript is whether the corrected Ne isotope data is indeed robust and whether this correction is valid. If this is the case, then the findings of this study may be of interest to the geoscience community.

We would like to take this opportunity to thank the reviewer for their detailed and helpful comments, and appreciate that they recognise the significance of our work. We are pleased to report that we have both examined the errors associated with our previous measurements and undertaken multiple re-analysis of the most mantle-rich sample that we had remaining and these confirm the presence of plume sourced Neon, removing the need for the correction procedure outlined in the previous version of the manuscript.

The previous analytical errors were governed by the reproducibility of air calibrations. We have now re-calculated the reproducibility of the calibrations ($n = 14$) by finding the best Gaussian fit to the probability density distribution of $^{21}\text{Ne}/^{22}\text{Ne}$ and $^{20}\text{Ne}/^{22}\text{Ne}$ ratios, which is an objective way of filtering outliers, improving the errors on the original data. These improved errors mean that the original Baker Farm sample plots more clearly off the established air-MORB line.

Furthermore, the reanalysis of the two most mantle rich sample, on a new instrument dedicated to high precision Ne measurements, allows constraint of the effect that $^{20}\text{NeH}^+$ has on the measured $^{21}\text{Ne}/^{22}\text{Ne}$ ratio. Correction for this contribution results in the new high precision samples plotting further off the Air-MORB line and makes our identification of plume sourced Ne in the samples to be considerably more robust.

On a more general note, the application of Ne isotopes to identifying the origin of mantle plume is not novel and has been widely used previously. Thus, I don't think the approach used in this study is novel enough and it would fit better in a more focused journal.

Whilst the application of Ne isotopes to identify the origin of a mantle plume is indeed not particularly novel, to date very few studies have been able to obtain Ne isotope measurements that are uncontaminated by air from natural CO_2 seeps. Furthermore, despite the presence of CO_2 degassing in the southern KwaZulu-Natal region, along with the variety of enigmatic geological phenomenon that we outline, geochemical evidence for the source of the degassed CO_2 has been

lacking due to the inaccessibility of volcanic products for fluid sampling and the difficulty of obtaining geochemical fingerprints that are uncontaminated from both air and radiogenic overprints in fluid samples. We believe that our findings are consistent with the presence of a plume component, similar to that of the Réunion or Kerguelen source, confirming the presence of the previous hypothesised Quathlamba mantle plume.

Validity of the Ne correction :

First, the authors have given some equations for the $3\text{He}/4\text{He}$ $\text{CO}_2/3\text{He}$ isotope data to identify various components but do not explain properly how the correction is done for Ne isotopes. This should be part of the manuscript. Why is the minimum contribution from radiogenic derived ^{21}Ne estimated by taking the difference between SCLM and the least radiogenic helium sample? This seems to make the assumption that there is a component with a $3\text{He}/4\text{He}$ ratio higher than 4.25 Ra in the mixture but are we certain this is the case ? If the ratio of the other component is lower than that of average SCLM, what happens to the corrected Ne isotope ? There are some reservoirs including mantle plumes characterized by low $3\text{He}/4\text{He}$. By making this assumption, the authors may have automatically produced a Ne reservoir with a $^{20}\text{Ne}/^{22}\text{Ne}$ versus $^{21}\text{Ne}/^{22}\text{Ne}$ signature typical of mantle plumes. In the absence of more detailed explanations, it seems that the corrected Ne data may not be so robust.

It is clear from the reviews received that we did not sufficiently explain both the procedure and the assumptions that we used to correct the Ne data for the presence of crustal contributions. However, as we now outline in detail, subsequent duplicate reanalysis of the most mantle rich samples – Mjaja and Baker Farm, now negates the need for such a correction, as both sets of duplicate analysis now plot outside of the air-MORB mixing line, implying a clear plume source of Ne.

The new analysis procedures and result are now outlined in the revised version of the manuscript and we invite the reviewer to examine this new data and determine if this addresses their concerns on the robustness of the identification of a plume source of Ne in the sampled CO_2 .

Second, there should be a propagation of error calculations for the corrected Ne data. The data shown in Figure 4a has relatively large errors and it is not clear whether propagation of errors has been done. There are uncertainties in the $4\text{He}/^{21}\text{Ne}$ ratios due to crustal production as these ratios depend on cross sections of neutron reactions with ^{24}Mg . Similarly the $3\text{He}/4\text{He}$ value of the SCLM has a typical variability of approximately 1 Ra.

As per our response to the comment above, it is clear that the errors associated with our correction of Ne isotopes for crustal contributions should have been better explained in the original manuscript. However, as this correction is now not required following reanalysis of the most mantle-rich sample available, this error propagation is no longer needed. Please see the revised version of the manuscript along with the response to Reviewer 3 on page 4 for further explanation of the new analyses that have been undertaken.

Specific Comments

Line 42 : Edge driven convection instead of edge driven convention

Same in line 48 'mantle convention'

We thank the reviewer for spotting these autocorrect errors – both are now corrected in the revised manuscript.

Line 57 Rifting and mantle plumes could be both related to each other. This idea has been previously proposed in the literature.

Indeed, we agree with the reviewer although this text has now been removed in the revised version of the manuscript.

Line 68 What is a moderately undegassed mantle ? I could understand the concept of a moderately degassed mantle but what a moderately undegassed mantle means is more obscure.

We thank the reviewer for this comment, and have now revised the text as follows to clarify our intended meaning.

Here we show that natural CO₂ seeps along the Ntlakwe-Bongwan fault within KwaZulu-Natal, South Africa, have C-He isotope systematics that support an origin from degassing mantle melts. Neon isotopes indicate that the melts originate from a deep mantle source that is similar to the mantle plume beneath Réunion Island, rather than the convecting upper mantle or sub-continental lithosphere.

Line 117 correct 'primoidal 3He'

Now corrected and we thank the reviewer for highlighting this typographical error.

Figure 3 : The ternary diagram used for representing the data in Figure is not a classical tool of noble gas geochemistry. The coordinates that are used are non linear and the components do not sum up to 100%. It would be good to explain perhaps in the supplement how this is built and how the coordinates are calculated. There is no sign of a mantle plume component in this diagram and this should be explained.

This form of ternary plot was first used by Giggenbach et al., 1993, GCA 57, p3427-3455 ([https://doi.org/10.1016/0016-7037\(93\)90549-C](https://doi.org/10.1016/0016-7037(93)90549-C)). It has subsequently been used many times as a means to resolve CO₂ origins, notably by the late David Hilton and his group. Previous examples of its use include Hilton et al., 1998 ([https://doi.org/10.1016/S0009-2541\(98\)00044-8](https://doi.org/10.1016/S0009-2541(98)00044-8)), Mutlu et al, 2008 (<https://doi.org/10.1016/j.chemgeo.2007.10.021>), de Leeuw et al 2010 (<https://doi.org/10.1016/j.apgeochem.2010.01.010>), Barry et al 2013 (<https://doi.org/10.1016/j.chemgeo.2012.07.003>). Hence, whilst it could be argued that it is not a "classic" noble gas diagram, we believe that it is an established diagram that is a very useful tool for resolving sources of CO₂, through combining CO₂^βHe, CO₂^αHe and ³He/⁴He on a single diagram.

Our form of presentation of this diagram follows the form taken in the papers listed above, but we agree that the figure would benefit from more precise labelling of the axes due to its relative complexity. Hence, we have now revised the figure to follow the example of Hilton et al 1998 in how the axis are labelled, which now shows the scaling values used. We have also added additional text to the figure caption to describe in more detail how these plots are created and highlighting that mixing and addition / loss do indeed plot as straight lines.

The discussion after line 214 is only a review of previous work and is not specifically original to this study. It is true that the presence of a mantle plume beneath south Africa forces one to reinterpret the observations made in this area. Yet the hypothesis of a mantle is not novel, which means this part of the interpretations does not make this study stronger. One other weakness is that the noble gas signature was only measured in gas seeps and therefore the connection with a mantle plume cannot be made for magmatic rocks directly.

We agree that this discussion provides a review of previous work, but we believe that this is useful to the reader in light of our confirmation of the mantle origin of the CO₂ seeps in the area. As we also provide the first confirmation of the presence of a plume source within the noble gas composition of the CO₂ degassing in the region, our results confirm that the enigmatic phenomenon present in the area can be accounted for by the early stage effects of the plume arriving underneath the region. Hence, we believe that the inclusion of this text provides an essential context to the study and how these relate to the geological setting of the region.

We acknowledge that we have only measured the noble gas signature in the CO₂ seeps and not the magmatic rocks in the area, meaning that the link between the two is not direct. However, as we outline in the manuscript, to date there has only been minor magmatic activity in the area and whilst we cannot directly confirm that this is directly related to the presence of the mantle plume underneath the region, it is the simplest explanation for the magmatic activity that has occurred. Furthermore this small-scale magmatic activity links to the other enigmatic geological features of the region that include active seismicity, elevated topography, an elevated geothermal gradient and thermal springs.

Reviewer #3 (Remarks to the Author):

This paper reports the first noble gas data from the Natal province in SA. Because these elements are powerful tracers of mantle-crust interaction and mantle domain provenance, this study is welcome to document an important geodynamical problem from a different perspective. The region presents several indications of recent tectonic and volcanic activities, and current models for this province advocate either the interaction of a mantle plume source with the cratonic lithosphere, and/or regional rifting.

The present data are not many, supposedly because the gas manifestations are scarce, and the authors did a good job in documenting the composition of several key geochemical tracers. They show without ambiguity the contribution of a mantle source for these CO₂-rich manifestations, which could not be demonstrated clearly with only stable isotope and chemistry data. The C/³He ratios of the samples with elevated ⁴⁰Ar/³⁶Ar ratios are within the range of mantle values, and the ³He/⁴He ratios values up to 4Ra cannot be a crustal signature.

We thank the reviewer for recognising the importance of our new noble gas data and the new approach to complement the existing geophysical imaging and modelling studies beneath southern South Africa. The reviewer is correct in their assumption that the gas manifestations are scarce, and we appreciate their kind words on our effective use of the different geochemical tracing tools to resolve a mantle origin for the CO₂ degassing at the springs.

In a further step, the authors argue that the mantle source is different from the convective mantle feeding mid-ocean ridges because the Ne isotope ratios lie above the air-MORB mixing line. Here the message is more ambiguous, because only one sample is marginally above the correlation, at 1 sigma level. Such measurements are generally difficult, but with modern instrumentation and the large quantity of sample allowed by natural gases, it should be possible to reach a better analytical precision and give the data at the 2 sigma level, potentially resolving this ambiguity.

We agree with the reviewer on this point, but the scarcity of the seeps and logistical challenge of sample collection unfortunately limits the volume of gas that we have available for reanalysis. We are pleased to report that we have both examined the errors associated with our previous measurements and undertaken multiple re-analysis of the most mantle-rich sample that we had

remaining and these confirm the presence of plume sourced Neon, removing the need for the correction procedure outlined in the previous version of the manuscript.

The previous analytical errors were governed by the reproducibility of air calibrations. We have now re-calculated the reproducibility of the calibrations ($n = 14$) by finding the best Gaussian fit to the probability density distribution of $^{21}\text{Ne}/^{22}\text{Ne}$ and $^{20}\text{Ne}/^{22}\text{Ne}$ ratios, which is an objective way of filtering outliers, improving the errors on the original data. These improved errors mean that the original Baker Farm sample plots more clearly off the established air-MORB line.

As we now outline in the revised manuscript, ongoing development at SUERC has resulted in a series of experiments that have determined how NeH^+ is formed in a low resolution static vacuum mass spectrometer with a standard Nier-type ion source. This work has resulted in the development of a new protocol to quantify the production of $^{20}\text{NeH}^+$ on the basis of the measured $^{22}\text{NeH}^+$, meaning that low resolution instruments can be used for accurate and precise Ne isotope determinations (see attached paper currently in revision following review with *Geochimica et Cosmochimica Acta* for further details).

Using this new procedure, we have undertaken duplicate reanalyse of the two mantle-rich samples, Baker Farm and Mjaja which we fortunately had remaining samples of. The results of this reanalysis are provided in the plot below, showing that both duplicate pairs of the reanalysed samples have significantly lower errors and both samples plot within the improved error margin of each other. Furthermore, due to the correction of the contribution of NeH^+ to both the $^{20}\text{Ne}/^{22}\text{Ne}$ and $^{21}\text{Ne}/^{22}\text{Ne}$, and the improved errors produced by the improved analysis precision, both duplicate sets of samples now plot outside of the Air-MORB line. Hence, and as highlighted in our response to reviewer 2, we believe that the duplicate reanalysis of the Mjaja and Baker Farm samples now negates the need for any correction to the Ne data to show that there is a clear plume source of Ne present in both samples.

The new analysis is summarised along the new results in the revised version of the manuscript and we invite the reviewer to examine this new data and determine if this addresses their concerns on the robustness of the identification of a plume source of Ne in the CO₂ sampled.

The authors then correct the data with the following assumptions: noble gases were contaminated by a crustal component, so that the He isotope ratio of 4Ra results from mixing of crustal radiogenic 4He with a local mantle component having a value within 4.9-7.1Ra, based on data from the Karoo large igneous province and on "nearby" volcanic islands (Comores, Tristan, Gough). I guess the concept of nearby may have a different flavour at the African continent scale, but the justification of this mantle end-member signature appears disputable. The authors argue that this is consistent with a subcontinental lithospheric mantle signature, in which case this appears difficult to reconcile with a less degassed mantle plume origin. Accepting this mantle He isotope signature, the authors propose to correct the Ne data for crustal contamination using the 4He/21Ne radiogenic production (or accumulation, not specified) ratio published in the literature. The ms or its supplementary should absolutely state the assumptions, detail the calculations, give the mixing equations for the 2 He isotopes and the 3 neon isotopes, and give the propagation of analytical errors and uncertainties on end-member compositions. Another concern is the assumption that crustal 4He and 21Ne are not fractionated during crustal contribution, which remains to be demonstrated. The authors could make some tests demonstrating the effect of potential fractionation on the corrected Ne isotope ratios.

Once more we thank the reviewer for these insightful comments, which are valid as we clearly did not explain the procedure for correcting the crustal contributions to the Ne isotopic ratios in sufficient detail or clarity. However, as outlined in detail above, our duplicate reanalysis of the two mantle rich samples, Baker Farm and Mjaja, now negates the need for any correction to the data.

Reviewer #4 (Remarks to the Author):

Gilfiligan et al. - Upwelling of the Quathalamba mantle plume beneath Southern Africa

This study uses noble gases to identify a mantle component in CO₂ gases collected from seeps in Southern Africa. The conclusions are justified and I believe the work is of wide interest. It is generally well written. I have only have minor comments some of which are stylistic. I recommend it is suitable for publication in Nature Communications with only minor modifications.

We would like to thank Mark Kendrick for recognising the novelty and interest of our work to the Geoscience community. We would also like to take this opportunity to thank him for a thorough and constructive review, which has significantly improved the manuscript. We have addressed the stylistic comments highlighted in his review as outlined below.

1)

The intro and abstract could be improved by stating more simply what observations need to be explained, why it is important and what has prevented progress previously. The last of these 3 ingredients is currently missing.

For example, the first line states 'The effect of upwelling mantle plumes on initiating uplift of cratonic continental crust and generating intraplate volcanism is poorly understood.'
This is too complex! There is a lot to unpack in that sentence which has already invoked mantle plumes. Nothing in the new data actually prove a link between plumes and initiation of uplift, so it is

not all that pertinent.

I think something simpler along the lines of 'Mantle plumes are sometimes invoked to explain the anomalously high topography and heatflow of southern Africa. However, geochemical evidence for the presence of a mantle plume or the nature of the mantle reservoir sampled has been lacking because....? (inaccessibility of volcanic products? ambiguous radiogenic isotope signatures? difficulting of attributing radiogenic isotopes to mantle source versus crustal contamination? why is the involvement of a plume ambiguous - I would like to know?). Here we use noble gases in CO₂ seeps to overcome these difficulties and show.....

We thank the reviewer for his very helpful and constructive comments on our abstract and introduction. We have taken these on board in the revised version the abstract and introduction which have been rewritten to focus on the link to the mantle underplating in the southern African region, and the current lack of direct geochemical evidence of mantle upwelling.

2)

Some of the language used to convey the significance of the work is a tad strong for my taste e.g. on line 19 - 'uniquely show an unambiguous..' the next sentence goes on to suggest the data characterise a primitive plume component rather than depleted mantle. The language may be technically correct, but the sentiment conveyed is a little strong. The evidence for mantle involvement is compelling, but the plume signature is not as compelling. I think it would be fairer to use words like 'favour a plume component' or are 'consistent with....'..a source similar to the Kerguelen source'.

We thank the reviewer for this helpful comment. We have revised the manuscript accordingly to tone down the language, and provide clarity to the reader on our sentiment. The line highlighted has been revised as follows:

Helium, neon and argon isotope systematics are consistent with the presence of a plume component, similar to that of the Réunion and Kerguelen source.

3)

Line 113 and thereabouts. What is the concentration of noble gases within the CO₂ gases?

The measured concentration values have now been added to the revised manuscript as follows:

⁴He exhibits the widest range in concentration, from 1.89 x 10⁻⁹ (Umtamvuna Mound 2) to 9.62 x 10⁻⁵ cm³(STP)cm⁻³ (Baker Farm). ²⁰Ne concentrations range from 4.01 x 10⁻⁹ to 4.13 x 10⁻⁸ cm³(STP)cm⁻³, with ⁴⁰Ar ranging from 6.34 x 10⁻⁶ to 7.83 x 10⁻⁵ cm³(STP)cm⁻³. As with ⁴He concentrations, the lowest ²⁰Ne and ⁴⁰Ar values are exhibited by the CO₂ sampled from Umtamvuna Mound 2, and the highest values are from the Baker Farm sample.

4)

I find Figure 3 confusing. This kind of trigonal plot is usually used to separate 3 end-members, in this case CO₂, ³He, ⁴He. However, the area shown represents a small fraction of the complete triangle - in this case the bottom axis still has 10,000 times more CO₂ than ⁴He (or put another way, the apices should be labelled as mixtures of gas, not pure ³He and ⁴He).

As per our comments to reviewer 2 (page 3) We agree that this is a complex diagram, but it is a very useful and established means of separating out different CO₂ sources by combining CO₂/³He, CO₂/⁴He and ³He/⁴He ratios on a single plot (see comments to reviewer 2 on this for further details of the origin and previous use of such plots).

However, we agree that the plot should have been better explained to the reader in the previous version of the manuscript, so we have included a more thorough explanation in the revised manuscript, both in the main text and in the figure caption.

Our labelling of the apices on this figure followed recent published versions of this diagram, but we agree with the reviewer that this over-simplified labelling of the apices introduces confusion. Therefore we have adopted the apex labelling style used by Hilton et al, 1998, which shows the scaling factors used to plot the diagram. We have also now added additional explanation of the plot in the main text and revised the figure caption to better explain the plot to the reader as follows:

Fig. 3: Ternary plot of $\text{CO}_2/{}^3\text{He}$, $\text{CO}_2/{}^4\text{He}$ and ${}^3\text{He}/{}^4\text{He}$ of the Bongwan CO_2 samples and relevant end-members, after Giggenbach et al, 1993. Relative concentrations are calculated for CO_2 , ${}^3\text{He}$, and ${}^4\text{He}$, based on these ratios. Scaling factors are introduced to the CO_2 and ${}^3\text{He}$ values, as labelled, to allow the spread of the data to be represented. The scaled relative concentrations are then normalised to a three-component system where $\text{CO}_2 + {}^3\text{He} + {}^4\text{He} = 100\%$, permitting direct comparison of the relative proportions of CO_2 , ${}^3\text{He}$, and ${}^4\text{He}$, regardless of absolute concentrations. Binary mixtures and loss or gain of any single component will plot as straight lines.

I am also unsure if the reference fields or axes are correctly labelled. Air is placed on the bottom axis at ${}^3\text{He}/{}^4\text{He}$ of 1 Ra. This should be a $\text{CO}_2/{}^4\text{He}$ of ~ 80 because air contains $\sim 5\text{ppm}$ ${}^4\text{He}$ and $\sim 400\text{ppm}$ CO_2 . It is difficult to read the $\text{CO}_2/{}^4\text{He}$ from the axis which has been raised $\times 10,000$, however, 80 divided by $10,000$ would be 0.0080 .

This analysis is correct and is consistent with our diagram. Importantly, the scaled concentrations of CO_2 , ${}^3\text{He}$, and ${}^4\text{He}$ are all normalised to 100% in a 3-component system, which we now describe more thoroughly (see above). This means that the CO_2 content of air plots at 0.01% of the height of our CO_2 axis, which is approximately equal to the thickness of the line defining the x-axis, and is thus visually indistinguishable.

Silicate rocks frequently contain hundreds of ppm range of CO_2 but are generally strongly enriched in ${}^4\text{He}$ compared to air and also the mantle. Therefore crust should surely have a $\text{CO}_2/{}^4\text{He}$ of less than air and less than the mantle?

The field labelled as crust might represent a specific unit or carbonate sediments, but I cannot foresee it is representative of the whole crust.

The reviewer is correct in that silicate rocks contain little CO_2 but relatively large quantities of ${}^4\text{He}$, which is the reason that we highlight that interaction of acidic groundwaters with carbonate rocks is the most likely source of the elevated $\text{CO}_2/{}^4\text{He}$ ratios we observe. Acidic waters would cause significant degassing of CO_2 from any carbonates that they encountered and significantly dilute any ${}^4\text{He}$ present in these sediments.

However, we thank the reviewer for pointing out our imprecise labelling. This field should have been labelled "crustal CO_2 " to reflect the source being carbonate sediments. This field plots ${}^3\text{He}/{}^4\text{He}$ ratios of 0.05, and $\text{CO}_2/{}^3\text{He}$ ratios between 1×10^{12} and 1×10^{14} , as used in Fig. 2. This has now been corrected in the new figure within the revised version of the manuscript.

As a result of the chosen fields one of the data points lies outside the apparently allowed area of mixing (shaded grey) and is plotted on a vector toward pure ${}^4\text{He}$. However, surely this vector is toward crust? (It's not pure ${}^4\text{He}$ given the apex is a ${}^4\text{He}$ -rich mixture....)

The CO₂ apex of the plot has been “magnified” using scaling factors, rather than plotting normalised absolute concentrations and then “zooming in” on the area of the plot of interest. This means that pure end members do not plot at the apices of this plot. As a result, the vector on the plot showing addition of ⁴He is correct.

The data appear to define a pseudo-binary array. I think the first end-member - Baker Farm (A) is a mixture of mantle and deep crustal gases (e.g. ancient silicates) and the 2nd endmember near the CO₂ apex is a ground water or carbonate-related component. This is similar to what is written, but I cannot reconcile what is written with the diagram - sorry!

Can the range of CO₂/⁴He in groundwater be plotted on this figure?

We agree with the interpretation of the reviewer in that the figure shows a pseudo-binary array. The Baker Farm endmember is indeed a mixture of mantle and deep sourced crustal gases, and the CO₂ endmember is the carbonate related component, which is much richer in CO₂ than ⁴He and ³He, resulting in addition of CO₂ with a low ⁴He, and virtually zero ³He content. As outlined in the manuscript this endmember could be accounted for by dissolution of carbonates by acidic groundwaters, effectively meaning that the potential CO₂/⁴He within the groundwaters is the same as that of the crust.

However, recent work that we have undertaken on similar CO₂ seeps in Australia has highlighted that solubility fractionation during exsolution of the CO₂ from the groundwater at CO₂ seeps can result in depleted ³He concentrations and associated high CO₂/³He which are commonly interpreted as evidence for mixing with crustal CO₂ (see - <https://doi.org/10.1016/j.gca.2019.06.002>). This means that the range of potential CO₂/⁴He ratios in from CO₂ seeps effectively has no maximum value, as it will depend on the degree of degassing of the groundwater that has occurred.

Based on this new knowledge we have revised the text on this section to clarify that the samples that lie on the mixing line towards the CO₂ endmember on figure 3 can be accounted for by the addition of CO₂ free of ³He and low in ⁴He and the effect of shallow preferential degassing of He relative to CO₂. As identification of the mantle end-member is the focus of this work, we do not see the benefit to the reader for further discussion on the origin of this CO₂ rich end-member, particularly as the small number of samples make it difficult to constrain the different contributions from mixing and shallow degassing processes.

5)

lines 168-169 - air equilibrated groundwater is mentioned here as a component so should be shown in Fig 3.

ASW can provide a significant source of Ar and Ne, but not for He and CO₂, as their concentrations in air are relatively low, compared to those of crustal and mantle fluids. Plotting ASW on Figure 3 would thus introduce an end-member that has no real bearing on the interpretation, and thus be more confusing for the reader. Given that two reviewers have highlighted that this figure is complex already, a point that we now believe we have addressed, we are reluctant to add another end member to the figure, particularly as we do not believe it is relevant to the data.

6)

Fig 4 is also confusing. There are six blue data points and 2 red ones. The red ones are discussed as being the measured values of the two samples with mantle Ne A and B (ref to Fig 3) and the blue points labelled A' and B' are corrected for radiogenic ingrowth.

Presumably the other blue points correspond to the measured values of the gases labelled C in Fig 3, but this needs to be clarified. Why aren't corresponding red points shown for these analyses? because there is no correction probably, but then why are the points blue instead of red?

This is indeed because there was no correction for those samples as they were close to the air value, with no real mantle contributions. Following the reanalysis these corrections are no longer needed and we hope that the reviewer finds the new plots to be clearer.

The best fit lines are not clear. Presumably they are forced through air? but this isn't stated. I don't think they are needed as we can already see the blue points lie close to the Kerguelen reference line.

The best fit lines from the original data are not forced through air, and this is now been clarified in the figure caption in the revised manuscript. As outlined previously, the correction previously applied is no longer required and it is now clear that the Baker Farm sample plots between the Kerguelen and Reunion reference lines.

Where does Ne from the Karoo province or Tristan de Cunha plot in this diagram? this would be more relevant than Kerguelen, as you mention it is a possible composition for background mantle when discussing Ne.

To our knowledge no Ne isotopic data from the Karoo province currently exists, with the only helium isotopic composition for the Karoo continental flood basalts or related rocks having been recently documented by Heinonen and Kurz in 2015 (<https://doi.org/10.1016/j.epsl.2015.06.030>). They attribute the lack of earlier data due to the scarcity of primitive samples with fresh olivine or glass containing primary magmatic gas inclusions that would be suitable for He measurements, along with the Jurassic age of the samples, and these issues make Ne analysis an even greater challenge, so it is unsurprising that no data currently exists.

Similar issues apply to Tristan de Cunha, as we are not aware of any reliable published Ne isotope data from this area either. The lack of available Ne data means that we cannot add to the existing reference lines on this plot.

Fig 4b doesn't really tell us anything useful. The slopes of the curves (not lines) are related to ^{36}Ar and ^{22}Ne concentrations in the end-members and could be altered by a number of processes not related to source composition. I would recommend moving this to the supplement as it doesn't help the nice story.

On reflection we agree with this comment, and given that the figure doesn't add anything useful we have removed it entirely from the manuscript.

7)

Methods. I appreciate space restrictions and that the methods have been described before and SUERC are experts in these techniques, but I think you should at least specify the instruments used for the measurements here. It wasn't all done on a quadrupole!!

We thank the reviewer for this valid observation. We have expanded the methods section to explain what instruments were used in the analysis and to expand on how the reanalysis of the Baker Farm sample has been performed.

Reviewers' comments:

Reviewer #2 (Remarks to the Author):

This manuscript has greatly improved with the addition of two new data points that seem to confirm a mantle origin for the CO₂ fluid emanation and the authors have adequately addressed my concerns. This means, as pointed out by the authors that a correction on the Ne isotope data is no longer required and the associated analytical uncertainties of the new data points are sufficient to demonstrate a mantle plume origin. With a more robust data set, I found this regional study is less ambiguous than in the original version and that it could have an interest for understanding the geodynamics of Africa. This is an uncommon superplume with no sign of magmatic activity at the surface but this result may be interesting in itself.

I only have a few minor comments to make :

Line 97 CO₂-rich rather than CO₂-rch

Line 100 more than one reference would be desirable.

Line 140 explain somewhere how the depth of mixing is inferred.

Figure 2 : Please note that the samples do not fall on the broad mixing line between mantle and carbonate.

Reviewer #5 (Remarks to the Author):

Review of "Upwelling of the Quathlamba mantle plume beneath southern South Africa"

Gilfillan et al present new He, Ne and Ar isotopic data from a series of natural CO₂ seeps in South Africa. The authors identify a primordial Ne isotopic signature, which they attribute to the upwelling of undegassed primitive mantle volatiles from the previously disputed Quathlamba mantle plume. The data are of high quality and the paper is well written. It represents a new advance in our understanding of the mantle beneath South Africa and therefore should be published if the authors can respond to my minor concerns.

Unlike the other authors this is my first time seeing this paper. I can see that it has been comprehensively reviewed by a the previous group of reviewers so I will not comment too much on the meat of the paper but focus on the modifications the authors have made and their responses to the previous reviewers concerns.

Major comments

I commend the authors for taking the time to reanalyse their samples using a more modern mass spectrometer in order to achieve a higher degree of precision than they were previously able to. I would however appreciate if the authors could add some more detail to their methodology section on how these new data were obtained.

The data analysed by the Argus mass spectrometer represent a considerable improvement over the previous analyses. I understand the authors refer to the newly published (congrats) paper by Gyôre et al., (2019), however I am of the opinion that for a high impact journal such as Nature Communications the reader (and myself) should be able to understand how the authors reached such a high level of precision without having to read another paper. Simply adding the number, and reproducibility, of standards run during the analysis would be a start. Also given the low resolution of the Argus can the authors outline how the $^{40}\text{Ar}^{++}$ contribution was monitored / corrected and how does this contribute to their errors i.e. what is the variability of $^{40}\text{Ar}^{+}/^{40}\text{Ar}^{++}$ ratios across different samples and standards. Potentially also consider adding the conditions of the source.

Continuing with the method section, I am a bit confused by the correction of $^{20}\text{NeH}^{+}$. Is this correction significant? Can the authors give a percentage? Potentially in the supplemental materials they can add a figure showing where the uncorrected values would plot on figure 4a. The reason I bring this up is I wonder whether the uncorrected values would fall on the MORB-air mixing line. If this is the case, doesn't that mean that all previously measured MORB values would now plot off and to the left of the MORB-air mixing line if they are corrected for $^{20}\text{NeH}^{+}$ contributions? Without outlining these corrections in more detail I would have doubts about whether these new Ne isotopic ratios are really showing a deep mantle contribution.

My only other major comment is the general lack of discussion concerning why the deep mantle input is only seen in the Ne isotopes. This is not a necessarily a deal breaker as noble gases are often shown to be decoupled from each other, for instance He from Ne in Iceland (Fûri et al., 2010; GCA) and He from Xe in Eifel (Caracausi et al., 2016; Nature). However, it would be nice to see some more interpretation from the authors as to why they think this is the case in these sites. For instance in Figure 2, why do $^3\text{He}/^4\text{He}$ ratios not correlate with mantle input? All values seem to have roughly the same $^3\text{He}/^4\text{He}$ ratios regardless of where they sit on the mantle-crust mixing line. Again these are not major concerns but I think a few more lines of interpretation of the data would significantly improve the paper.

Minor Comments

Several authors are listed as being at institution number 6, which does not appear to exist.

Line 17 - "High heat flow" is a bit vague.

Line 63 - Although commonly used to identify plume inputs, noble gases do not technically give information on the mantle depth, only the degassing state of the mantle source. Maybe add "between deep undegassed and shallow convective mantle sources".

Line 67 - Give the typical $^3\text{He}/^4\text{He}$ for primordial mantle plumes $>8R_a$.

Line 95 - Maybe put this paragraph before the previous one, so the concept of CO_2 degassing is introduced before specifically discussing southern Africa.

line 99 - Replace "impossible" with something less strong, maybe difficult or challenging.

Line 127 - Compared to what? The other noble gases.

Line 134 - Give the $^3\text{He}/^4\text{He}$ of the crust here.

Line 155 - Maybe replace "allowing" so you don't use allow twice in one sentence.

Line 157-158 - Give the percent of crustal contribution required to explain the Baker Farm data assuming it started with a $^3\text{He}/^4\text{He}$ similar to DM or SCLM.

Line 240 - There was very little discussion of the origin of Ar in the discussion and couldn't the high $^{40}\text{Ar}/^{36}\text{Ar}$ ratios be related to crust inputs. I think the only real constraint is from Ne isotopes so I would concentrate on them.

Response to previous reviewers concerns

The authors state that they improved the precision on their previous measurements by filtering the standards using a Gaussian fit but how many of the 14 standards were excluded?

Response to Reviewers' comments from second review round

Reviewer #2 (Remarks to the Author):

This manuscript has greatly improved with the addition of two new data points that seem to confirm a mantle origin for the CO₂ fluid emanation and the authors have adequately addressed my concerns. This means, as pointed out by the authors that a correction on the Ne isotope data is no longer required and the associated analytical uncertainties of the new data points are sufficient to demonstrate a mantle plume origin. With a more robust data set, I found this regional study is less ambiguous than in the original version and that it could have an interest for understanding the geodynamics of Africa. This is an uncommon superplume with no sign of magmatic activity at the surface but this result may be interesting in itself.

We would like to take this opportunity to again thank the reviewer for their time and helpful comments which have greatly improved the manuscript. We are pleased to hear that we have now addressed their main concerns and we outline below how we have now addressed the minor comments they have raised.

Line 97 CO₂-rich rather than CO₂-rch

- This typographical error is now corrected

Line 100 more than one reference would be desirable.

- *Additional references to the following works now added:*
- Trull, T., Nadeau, S., Pineau, F., Polve, M. & Javoy, M. C-He systematics in hotspot xenoliths: Implications for mantle carbon contents and carbon recycling. *Earth Planet. Sci. Lett.* **118**, 43–64 (1993).
- Burnard, P., Graham, D. & Turner, G. Vesicle-specific noble gas analysis of “popping rock”: Implications for primordial noble gases in earth. *Science* 276, 568–571 (1997).

Line 140 explain somewhere how the depth of mixing is inferred.

- *We derive this depth of mixing by using the average surface temperature of 14°C and the geothermal gradient of 30°C/km. This has now been clarified in the revised manuscript text.*

Figure 2: Please note that the samples do not fall on the broad mixing line between mantle and carbonate.

- *This is indeed the case, and we outline how the trend between CO₂/³He and δ¹³C_{CO2} are consistent with the mixing of mantle-derived CO₂ with CO₂ derived from the overlying Marble Delta Formation carbonates at up to 70°C.*

Reviewer #5 (Remarks to the Author):

Review of "Upwelling of the Quathlamba mantle plume beneath southern South Africa"

Gilfillan et al present new He, Ne and Ar isotopic data from a series of natural CO₂ seeps in South Africa. The authors identify a primordial Ne isotopic signature, which they attribute to the upwelling of undegassed primitive mantle volatiles from the previously disputed Quathlamba mantle plume. The data are of high quality and the paper is well written. It represents a new advance in our understanding of the mantle beneath South Africa and therefore should be published if the authors can respond to my minor concerns.

- *We would like to take this opportunity to thank the reviewer for their time in reviewing our manuscript and for their helpful and constructive comments. We outline our response to these in the following text.*

Unlike the other authors this is my first time seeing this paper. I can see that it has been comprehensively reviewed by a the previous group of reviewers so I will not comment too much on the meat of the paper but focus on the modifications the authors have made and their responses to the previous reviewers concerns.

- *We again thank the reviewer for this sensible approach to reviewing our work.*

Major comments

I commend the authors for taking the time to reanalyse their samples using a more modern mass spectrometer in order to achieve a higher degree of precision than they were previously able to. I would however appreciate if the authors could add some more detail to their methodology section on how these new data were obtained.

The data analysed by the Argus mass spectrometer represent a considerable improvement over the previous analyses. I understand the authors refer to the newly published (congrats) paper by Gyôre et al., (2019), however I am of the opinion that for a high impact journal such as Nature Communications the reader (and myself) should be able to understand how the authors reached such a high level of precision without having to read another paper. Simply adding the number, and reproducibility, of standards run during the analysis would be a start. Also given the low resolution of the Argus can the authors outline how the ⁴⁰Ar⁺⁺ contribution was monitored / corrected and how does this contribute to their errors i.e. what is the variability of ⁴⁰Ar⁺/⁴⁰Ar⁺⁺ ratios across different samples and standards. Potentially also consider adding the conditions of the source.

- *We agree with the reviewer on this point, but were previously limited to the fact that the paper outlining the new methodology had yet to be published. As the reviewer states this paper has now been published, removing this constraint. Hence we have now revised the methods section to outline how these high precision analysis was undertaken.*

Continuing with the method section, I am a bit confused by the correction of ²⁰NeH⁺. Is this correction significant? Can the authors give a percentage? Potentially in the supplemental materials they can add a figure showing where the uncorrected values would plot on figure 4a. The reason I bring this up is I wonder whether the uncorrected values would fall on the MORB-air mixing line. If this is the case, doesn't that mean that all previously measured MORB values would now plot off and to the left of the MORB-air mixing line if they are corrected for ²⁰NeH⁺ contributions? Without

outlining these corrections in more detail I would have doubts about whether these new Ne isotopic ratios are really showing a deep mantle contribution.

- As we now outline in the revised extended methods, the $^{20}\text{NeH}^+$ correction at $m/z = 21$ was found to be 0.96 % (Baker Farm) and 1.94 % (Mjaja). The contribution of NeH correction toward the overall uncertainty is for these two samples is +0.02 %. As suggested by the reviewer we have now added a plot to the supplementary information which shows the uncorrected high precision Ne analysis, along with the original low precision measurements (also shown below for convenience).

- Plot of $^{20}\text{Ne}/^{22}\text{Ne}$ against $^{21}\text{Ne}/^{22}\text{Ne}$ for all of the measured Bongwan CO_2 samples, including low precision uncorrected for NeH^+ (red diamonds); high precision uncorrected for NeH^+ (red diamonds) and the high precision corrected for NeH^+ (red triangles). This highlights that the Mjaja sample is within the error envelope of the low precision measurement, which was not corrected for the $^{20}\text{NeH}^+$ contribution to ^{21}Ne . The uncorrected high precision Baker Farm sample plots further above the air-MORB line than the low precision measurement. This can be explained by the fact that the high precision analysis was performed on a separate sample of the CO_2 from the Baker Farm well, which was collected at a later time, after the well had been sealed off from the atmosphere for a greater period of time. The approximately 1 hour period of magmatic gas accumulation in the sealed well between sample collection can account for the more mantle enriched signature of the uncorrected high precision sample.

My only other major comment is the general lack of discussion concerning why the deep mantle input is only seen in the Ne isotopes. This is not necessarily a deal breaker as noble gases are often shown to be decoupled from each other, for instance He from Ne in Iceland (Fûri et al., 2010; GCA) and He from Xe in Eifel (Caracausi et al., 2016; Nature). However, it would be nice to see some more interpretation from the authors as to why they think this is the case in these sites. For instance in Figure 2, why do $^3\text{He}/^4\text{He}$ ratios not correlate with mantle input? All values seem to have roughly the same $^3\text{He}/^4\text{He}$ ratios regardless of where they sit on the mantle-crust mixing line. Again these are not major concerns but I think a few more lines of interpretation of the data would significantly improve the paper.

- *We thank the reviewer for this suggestion and we have now added the following text to account for the reason that the mantle input is only observed in the Ne isotopes:*

Lines 217 to 224:

However, the $^3\text{He}/^4\text{He}$ of the sub-lithospheric mantle source of the nearby Karoo Large Igneous Province is cited as $7.03 \pm 0.23 R_A^{55}$ and the most proximal volcanic ocean islands (Comores, Tristan da Cuna and Gough) are characterised by low $^3\text{He}/^4\text{He}$ (4.9 to 7.1 R_A)⁵⁵. Hence, the mantle source in the region may have a $^3\text{He}/^4\text{He}$ that is not elevated above the MORB range ($8 \pm 1 R_A$), though is most likely to be above the highest measured $^3\text{He}/^4\text{He}_c$ of 4.5 R_A . It is also possible that the He and Ne systematics of the mantle under southern South Africa are decoupled, as has been observed in the Icelandic mantle source⁵⁶. This decoupling was attributed to either more compatible behaviour of He during low-degree partial melting or more extensive diffusive loss of He relative to the heavier noble gases. Incorporation of crustal-radiogenic ^{21}Ne to the Bongwan gases is also probable, but without constraint of the original Bongwan mantle $^3\text{He}/^4\text{He}$ this is impossible to determine.

- *Concerning Figure 2 and the lack of correlation between the variations in the mantle input reflected by the $\text{CO}_2/\beta\text{He}$ and constant $^3\text{He}/^4\text{He}$, this is due to the crustal CO_2 (lacking significant He) addition from groundwater and/or He loss due to degassing of CO_2 from groundwater, compared to the more mantle-rich $\text{CO}_2/\beta\text{He}$ of Mjaja and Baker Farm. This has now been clarified in the revised text (lines 207 - 209).*

Minor Comments

Several authors are listed as being at institution number 6, which does not appear to exist.

- *Now corrected, and we thank the reviewer for highlighting this error.*

Line 17 - "High heat flow" is a bit vague.

- *High now amended to anomolous – the word limit on the abstract limits further additions.*

Line 63 - Although commonly used to identify plume inputs, noble gases do not technically give information on the mantle depth, only the degassing state of the mantle source. Maybe add "between deep undegassed and shallow convective mantle sources".

- *Now amended as suggested.*

Line 67 - Give the typical $^3\text{He}/^4\text{He}$ for primordial mantle plumes $>8R_A$.

- *Now given as suggested.*

Line 95 - Maybe put this paragraph before the previous one, so the concept of CO₂ degassing is introduced before specifically discussing southern Africa.

- *We thank the reviewer for this suggestion, but we believe that the paragraph order is better as is, because this introduces the reader to the background carbon isotope measurements that have been made previously and then outlines the additional information on CO₂ origin that primordial noble gases can provide.*

line 99 - Replace "impossible" with something less strong, maybe difficult or challenging.

- *Now replaced with challenging as suggested.*

Line 127 - Compared to what?

- *'compared to the other noble gases' now added.*

Line 134 - Give the 3He/4He of the crust here.

- *Now added.*

Line 155 - Maybe replace "allowing" so you don't use allow twice in one sentence.

- *We thank the reviewer for this helpful suggestion and 'allowing' has now been amended to 'permitting'.*

Line 157-158 - Give the percent of crustal contribution required to explain the Baker Farm data assuming it started with a 3He/4He similar to DM or SCLM.

- *Now added as suggested.*

Line 240 - There was very little discussion of the origin of Ar in the discussion and couldn't the high 40Ar/36Ar ratios be related to crust inputs. I think the only real constraint is from Ne isotopes so I would concentrate on them.

- *On reflection we agree with the reviewer on this point and have removed the reference to Ar as suggested.*

Response to previous reviewers concerns

The authors state that they improved the precision on their previous measurements by filtering the standards using a Gaussian fit but how many of the 14 standards were excluded?

- *The Gaussian fit to the probability density distribution is an objective way to filter outliers. Consequently, there was no need to reject any of the 14 calibrations from this fit.*

Reviewers' comments:

Reviewer #5 (Remarks to the Author):

Review of "Upwelling of the Quathlamba mantle plume beneath southern South Africa"

In the previous version of the manuscript in general I was concerned with the lack of detail regarding the methodology used to perform the analysis. The authors have now outlined in more detail how the analyses were performed and as such, the manuscript is now much improved. I have few final concerns regarding the methods section but I am sure the authors can address these, at which point I would find the paper suitable for publication.

Minor Concerns

Overall the new method section is a vast improvement over the previous version. However, there are still a few details missing that I feel should be added. For the high precision analysis on the Argus, the authors state that the gas was expanded into a 2 litre steel volume and then multiple 100cm³ shots were taken and analysed. However, the authors do not state how many repeat analyses they undertook for each sample. Furthermore, I think it is vital for the raw data for each analysis to be presented in a table or figure in the supplements. Finally, since the authors analysed the sample multiple times did they calculate the error for each sample using the standard error on the distribution of the measurements? If the errors were calculated using standard error ((standard deviation/ $\sqrt{n-1}$) I wonder whether the multiple measurements can be considered truly independent repeat analyses since all the gas was pre-purified prior to being stored in the steel volume. In such case, any error introduced during the purification process would be systematic in all the subsequent measurements and not properly accounted for. At the very least it should be stated if the errors were calculated using standard errors and how many repeat analyses were performed for each sample. Of course these comments may not be applicable if the errors represent standard deviation.

Minor corrections

Line 307 - Should be ⁴⁰Ar⁺⁺, again on line 309

Line 308 - Should be CO₂⁺⁺, again on line 310

Line 316 - is the + necessary before the 0.02%?

I didn't seem to find any coordinates for the sampling sites; these should be added to one of the tables.

Michael Broadley

Response to Reviewer #5 – 16th September 2019

Compiled by Stuart Gilfillan on behalf of all co-authors

Reviewer #5 (Remarks to the Author):

Review of "Upwelling of the Quathlamba mantle plume beneath southern South Africa"

In the previous version of the manuscript in general I was concerned with the lack of detail regarding the methodology used to perform the analysis. The authors have now outlined in more detail how the analyses were performed and as such, the manuscript is now much improved. I have few final concerns regarding the methods section but I am sure the authors can address these, at which point I would find the paper suitable for publication.

- *We thank the reviewer once again for their time to review our revised manuscript. We are pleased to hear that they find that the manuscript has been much improved. We have now addressed the minor comments the reviewer has raised as outlined below.*

Minor Concerns

Overall the new method section is a vast improvement over the previous version. However, there are still a few details missing that I feel should be added. For the high precision analysis on the Argus, the authors state that the gas was expanded in to a 2 litre steel volume and then multiple 100cm³ shots were taken and analysed. However, the authors do not state how many repeat analyses they undertook for each sample. Furthermore, I think it is vital for the raw data for each analysis to be presented in a table or figure in the supplements.

- *This appears to be a misunderstanding by the reviewer. Each individual sample analysed on the ARGUS consisted of one 100 cm³ aliquot of the 2 litre of gas pre-cleaned on the sample preparation line attached to the MAP-215 mass spectrometer. This 100cm³ of gas was then further cleaned and admitted to the ARGUS mass spectrometer for analysis. A second aliquot of 100 cm³ of pre-cleaned gas from the same 2 L volume of pre-cleaned gas was then analysed as a duplicate measurement. This is standard practice for repeat analysis of the same sample across the entire of the noble gas community. However, it is obvious that this was not as clear as it could have been so we have now amended the text as follows:*

High precision analysis of Ne isotopes within the Baker Farm and Mjaja samples was undertaken in multi-collection mode using a ThermoFisher ARGUS VI using the following procedures. Each copper tube sample was mounted on the ultra-high vacuum line attached to the MAP 215-50 mass spectrometer, and subjected to the same clean up procedure as outlined above, following which they were trapped in a 2 L stainless steel cylinder. Approximately 100 cm³ of total gas was extracted from this cylinder to the ultra-high vacuum system attached to the ARGUS VI mass spectrometer as described in previous work⁶⁰. The gas was exposed to another SAES GP50 ZrAl getter (held at 250 °C) for 15 minutes and then a liquid nitrogen-cooled charcoal finger (held at -196 °C) for 15 minutes to trap any remaining active gases along with Ar, Kr and Xe. Ne

was then adsorbed on charcoal using an IceOxford cryopump (-243 °C, 20 min) while He was pumped away. Pure Ne was released at -173 °C and let into the ARGUS VI low resolution mass spectrometer. The clean-up and analysis procedure was undertaken twice for both the Baker Farm and Mjaja samples, and the results of the individual repeat measurements are plotted on Fig. 4 and Fig. SI-3, and listed in Table SI-2.

Finally, since the authors analysed the sample multiple times did they calculate the error for each sample using the standard error on the distribution of the measurements? If the errors were calculated using standard error ((standard deviation/ $\sqrt{(n-1)}$) I wonder whether the multiple measurements can be considered truly independent repeat analyses since all the gas was pre-purified prior to being stored in the steel volume. In such case, any error introduced during the purification process would be systematic in all the subsequent measurements and not properly accounted for. At the very least it should be stated if the errors were calculated using standard errors and how many repeat analyses were performed for each sample. Of course these comments may not be applicable if the errors represent standard deviation.

- As per our response to the comment above, this is a misunderstanding on the part of the reviewer over our analysis procedures. We outline the standard errors associated with our individual duplicate measurements in Table SI-2 and on Figs. 4 and SI-3. If such pre-purified measurements are not considered to be true duplicate measurements then this is news to us, as it is common practice across the noble gas community to repeat measurements on the same pre-cleaned gases to check consistency of analysis procedures eg. Ballentine et al., 2005, Nature, 433, p33-38 2005 (Sample BD12) and Holland et al., 2009 Science, 326, p1522-1525 – all samples.

Minor corrections

Line 307 - Should be 40Ar^{++} , again on line 309

Line 308 - Should be CO_2^{++} , again on line 310

Line 316 - is the + necessary before the 0.02%?

- These have now been corrected and we thank the reviewer for spotting these errors.

I didn't seem to find any coordinates for the sampling sites; these should be added to one of the tables.

- Another good spot, and again these have now been added to the revised manuscript.

REVIEWERS' COMMENTS:

Reviewer #5 (Remarks to the Author):

The authors have adequately addressed all of my concerns and I am now happy for the paper to be published in Nature Communication.